# Association of TP53 with Defective Long Chain 3-Hydroxy acyl-CoA Dehydrogenase Induced Non-Cirrhotic Hepatocellular Carcinoma

**DOI:** 10.3390/cancers17193241

**Published:** 2025-10-06

**Authors:** Tripti Khare, Alexei J. Stuckel, Suneel Gupta, Karina Liu, Ghassan M. Hammoud, Jamal A. Ibdah, Sharad Khare

**Affiliations:** 1Division of Gastroenterology and Hepatology, Department of Medicine, University of Missouri, Columbia, MO 65212, USA; kharet@health.missouri.edu (T.K.); hammoudg@health.missouri.edu (G.M.H.); ibdahj@health.missouri.edu (J.A.I.); 2Harry S Truman Veterans Hospital, Columbia, MO 65201, USA; 3MOgene, Welsch Industrial Ct, St. Louis, MO 63146, USA; astuckel@mogene.com; 4Department of Veterinary Medicine and Surgery, University of Missouri, Columbia, MO 65211, USA; guptasun@missouri.edu; 5Department of Biochemistry and Molecular Biology, Bloomberg School of Public Health, Johns Hopkins University, Baltimore, MD 21205, USA; kliu100@jh.edu

**Keywords:** MASLD, HCC, non-cirrhotic HCC, ingenuity pathway analysis

## Abstract

Researchers have studied how an energy process called fatty acid oxidation may contribute to liver cancer. They have used a unique mouse model that has a defective enzyme for fatty acid oxidation and found that some of the mice developed liver cancer without the typical hardening of tissue or scarring, or cirrhosis, which is usually present in liver cancer. This finding is important because it identifies a unique manner that liver cancer can start in some patients, which makes it harder to detect and treat patients using standard tools and methods. To address this challenge, the researchers used a protein analysis technique (proteomics) and software (Ingenuity Pathway Analysis) to identify specific molecular clues or markers. These results could lead to better strategies for identifying and treating liver cancer patients who do not have cirrhosis.

## 1. Introduction

Hepatocellular carcinoma (HCC), the third leading cause of cancer-related deaths worldwide, is the most common primary neoplasm of the liver [1,2]. It is more prevalent in developing countries. However, recently its incidence has increased in developed countries [3]. HCC typically develops in the setting of cirrhotic liver (HCC-C) but more recently about 20% of HCC cases have been reported in non-cirrhotic liver (HCC-NC), that gives rise to a serious challenge regarding cancer surveillance [4,5,6]. An unfamiliar variant of HCC, fibrolamellar carcinoma, also develops in patients with HCC-NC liver [7]. The incidence of HCC associated with hepatitis B and C viral infection recently has decreased in Japan and developed countries, but more non-viral cases of HCC have been reported in patients with metabolic dysfunction-associated steatotic liver disease (MASLD), including metabolic dysfunction-associated steatohepatitis (MASH) and related complications [8]. The association of 20–50% MASLD with HCC-NC is well documented in the recent literature related to human disease and experimental mouse models [9,10]. However, the surveillance and management of HCC in MASLD patients with no cirrhosis is hampered because of incomplete understanding of the mechanism of carcinogenesis and associated molecular events, and it is becoming a major public health concern [11]. The risk associated with HCC development in MASLD patients with no cirrhosis is much less than that in cirrhotic liver patients, and the annual incidence does not exceed the threshold for surveillance; that is why HCC screening is recommended only to patients with cirrhotic liver [12].

The etiology, cytogenetic, histopathologic, and clinical features of HCC-NC are very distinct, and patients have better disease-free survival than HCC-C patients despite the larger cancer burden at the time of diagnosis [13]. However, the angiogenic characteristics of HCC-C and HCC-NC are notably similar [14]. Recently, Jamwal et al. reported that HCC-C and HCC-NC have different clinic-radiological characteristics. HCC-NC is large, solitary, moderately differentiated, has well-defined margins, shows mosaic pattern of enhancement, complete capsule, and delayed phase washout, whereas HCC-C is small, multifocal, has higher levels of alpha fetoprotein, but poor differentiation and indistinct margins, heterogeneous enhancement, absent capsule, and portal venous phase washout [15]. So far, the molecular basis of differentiation between HCC-C and HCC-NC is poorly understood [16]. The aberrant methylation of promoter region of tumor-associated genes and DNA methylation markers with important diagnostic values were noticed in both HCC-C and HCC-NC [17,18]. In addition, RNA-seq data downloaded from The Cancer Genome Atlas (TCGA) database revealed differential expression of some key genes in both HCC-C and HCC-NC with significant diagnostic value [19].

We have described defective long-chain 3-hydroxyacyl-CoA dehydrogenase (LCHAD) expression as a new etiology associated with MASLD and non-cirrhotic HCC (HCC-NC) in a mouse model and provided an insight into the role of mitochondrial beta (β)-oxidation of fatty acids [9]. Mitochondrial β-oxidation of fatty acids is catalyzed by mitochondrial trifunctional protein (MTP), a hetero-octamer, composed of four α subunits (HADHA) and four β subunits (HADHB), catalyzing the last three steps of β-oxidation of long chain fatty acids [20,21]. We observed embryonic lethality in homozygous male mice with isolated LCHAD deficiency by deletion of exon 15 in the in-house developed mouse model (unpublished data), whereas heterozygous mice (HT) develop significant hepatic steatosis starting at young age (3 months old) and HCC at older age (~20% mice, 13 months or older) without any evidence of fibrosis or cirrhosis (HCC-NC). Remaining ~80% HT and WT mice never developed HCC. Thus, isolated LCHAD-defective HT mice had MASLD and developed hepatic tumors [9].

In the present communication, in order to analyze differences in biological networks, we utilized Ingenuity Pathway Analysis (IPA) and analyzed twenty-four differentially expressed proteins observed in heterozygous LCHAD mice without any sign of HCC (HT) when compared to wild-type (WT) mice [9]. The IPA transforms large data sets into biological networks with direct and indirect relationships between genes based on reported interactions in the literature [22]. The analyzed biological networks will thereby help in better understanding of the molecular mechanism(s) involved in the development of HCC-NC. Improved knowledge of molecular control of HCC-NC will help clinicians in taking better care of patients and detecting HCC-NC at early stages.

## 2. Materials and Methods

### 2.1. Study of cDNA Array of HCC Patients

Commercially available TissueScan Liver Cancer cDNA Array I (Cat # LVRT101) of Origene (Rockville, MD, USA) was utilized using the manufacturer’s protocol to assess HADHA transcript levels. The array plate had cDNAs from eight normal and twenty-six liver cancer tissues at different stages of carcinogenesis (seven stage I, eight stage II, and eleven stage III and IV) and different grades of tumor (six grade 1, thirteen grade 2, and four grade 3). As described by Origene, samples utilized in the study were identified by the pathologist based on the H&E slide, which reflects the tissue block before cDNA extraction. If cirrhosis is identified on the H&E slide, the pathologist incorporates this finding into the diagnosis. Eight samples were classified as “within normal limits” by the pathologist examining the H&E slide, with the composition characterized as “100% Normal”. Twenty-six samples were classified as “Cancers” with the diagnosis determined from the H&E slide. No cirrhosis was reported for the individual tissue samples stored. However, in the surgical pathology report, four patients were identified with cirrhosis in the background: one in stage I, two in stage II, and one in stage III. Consequently, most samples used in this study exhibited no cirrhotic background and no evidence of cirrhosis as determined by H&E staining. To measure HADHA transcripts, specific forward and reverse ready-to-use primers and a custom RT^2^ qPCR Primer Assay (cat. no. 330001, Qiagen, Germantown, MD, USA) was generated. Thermocycler (LightCycler^®^ 96-Roche, Madison, WI, USA) was used to perform PCR using the qSTAR SYBR Master Mix Kit (Bio-Rad, Hercules, CA, USA). The cycling conditions used were as follows: 3 min at 95 °C, followed by 50 cycles: 10 s at 95 °C, 30 s at 56 °C, 15 s at 72 °C, with an additional 7 min at 72 °C and then cooling down. Control reaction was performed in the absence of template for each primer pair for every PCR run. The manufacturer’s software was used to perform amplicon sizing and relative quantitation. Comparative 2^−ΔΔCt^ method was used to analyze qPCR data and HADHA transcript levels were normalized to β-actin and expressed as fold-control patients.

### 2.2. Ingenuity Pathway Analysis

Twenty-four proteins expressed differentially in heterozygous LCHAD mice with no cancer (HT) compared to wild-type (WT) mice, as reported in Appendix A in our earlier publication [9] or Appendix A in the present publication, were imported along with their log fold change values into Qiagen’s Ingenuity Pathway Analysis (IPA) software version 01-14 (01-14), and core analysis was conducted. IPA core analysis identifies a series of canonical pathways, diseases, and functions or networks relevant for the molecules in the input list. IPA core analysis uses a human-curated comprehensive knowledge base to provide valuable perspective and profound understanding of experimental datasets. Networks of potentially interacting proteins were identified and placed into node–edge diagrams with gene and gene products represented as nodes and biological relationship between two nodes as an edge (line). These biological networks are supported by at least one reference from the literature, from a textbook, or from ingenuity pathway knowledge base stored canonical information. Only networks with more than one focus molecules are presented. Each signaling network was provided with a P-score that factors in *p*-values [−log_10_(*p*-value)] generated by Fisher’s exact test; therefore, an increased P-score can be interpreted as a decrease in the odds of a network being generated at random. The hepatotoxicity report which is part of IPA’s core analysis under Tox functions was also generated. In the hepatotoxicity report, the twenty-four differentially expressed proteins were evaluated for enrichment of liver-related functional groups that surpassed the −log(*p*-value) threshold of 1.25 shown on the *Y*-axis of the bar chart.

### 2.3. SDS-PAGE and Western Blotting

Previously frozen liver tissues from 13-month-old or older WT mice, HT (heterozygous-LCHAD mice without cancer), and HT-NC (heterozygous-LCHAD mice with non-cirrhotic cancer) as described [9], were utilized. Frozen liver tissues were then lysed with RIPA buffer, and whole-cell lysates were collected. Protein concentration in the whole-cell lysates was determined using BCA protein assay kit (ThermoFisher Scientific-Pierce, Rockford, IL, USA). The SDS-PAGE electrophoresis was performed on individual liver cell lysates (30 µg per lane) by running on 10% Criterion-XT Bio-Rad gels [23], and proteins separated from the gel were transferred onto nitrocellulose 0.2 µm membranes (Bio-Rad, Hercules, CA, USA). The membranes were blocked with 5% BSA in Tris-buffered saline-tween20 (TBST) and further incubated overnight at 4 °C with diluted primary antibodies for p53 (1:1000), MDM2 (1:500) and βactin (1:2500) with constant shaking. After washing three times with TBST, the membranes were incubated with diluted horseradish peroxidase (HRP) labeled secondary antibody (1:5000) for 1.0 h at room temperature, washed with TBST several times, and signal was detected using the Super signal^®^ West femto maximum sensitivity substrate (Pierce). The images were visualized and quantified using ImageLab software 6.1 (BioRad, Hercules, CA, USA). Protein levels were expressed as relative expression/β-actin (mean ± SD). The results are presented as the mean ± SD of three independent samples. The primary antibodies for p53 (#282), MDM2 (#51541) and β-actin (#3700) and secondary antibodies, anti-mouse HRP-linked (#7076S), anti-rabbit HRP-linked (#7074S), were purchased from Cell Signaling Technology, Danvers, MA, USA.

### 2.4. Statistical Analysis

All statistical analyses were carried out using GraphPad Prism 10.2.0 software (GraphPad Software, La Jolla, CA, USA). For analyzing the statistical differences in mean HADHA transcript levels in control and HCC patients and Western blot data, one-way analysis of variance (one-way ANOVA) followed by a Bonferroni or Tukey post hoc multiple comparison test was used. The standard error of the mean (SEM) was used to express the average result values of data. The *p*-values ≤ 0.05, were considered statistically significant.

## 3. Results

### 3.1. HADHA Transcript Levels in cDNA Array of HCC Patients

In HCC, “disease status” describes the overall extent of the cancer including the size and number of tumors, extent of liver damage, and spread of cancer beyond liver, essentially indicating the stage of cancer, while “tumor grade” describes the aggressiveness of the cancer cells within the tumor itself, depending on how closely they resemble normal liver cells. More aggressive or higher-grade tumors are associated with more spread and poorly differentiated liver cancer cells. To understand the role of LCHAD, HADHA transcript level was measured in control and HCC patients with different stages and grades of tumor development. HADHA transcript levels normalized with β-actin in TissueScan Liver Cancer cDNA array from Origene were shown in Figure 1. We noticed a gradual and significant decrease in HADHA transcript levels based on disease status from stage 1 through 3/4 as compared to normal liver cells (*p* < 0.01 to *p* < 0.0001). However, based on tumor grade, the decrease in HADHA transcript level is not significant (ns, *p* = 0.999) between normal and grade 1 tumors but it is highly significant between normal and grade 2 (*p* < 0.0001) and 3 (*p* < 0.001) tumors.

### 3.2. Signaling Networks Associated with Differentially Expressed Proteins Between HT (Heterozygous LCHAD Mice Without Cancer) and WT Mice

The use of Qiagen’s IPA software and core analysis conducted on the 24 differentially expressed proteins between HT-LCHAD without cancer and WT mice (Appendix A) identified two statistically and biologically significant signaling networks, each containing 35 nodes associated with the 24 proteins of interest (Figure 2 and Figure 3). Under Top Diseases and Functions, network 1 consisted of 35 nodes and included a total of 11 differentially expressed proteins (10 downregulated and 1 upregulated) with a P-score = 25 that were associated with three affected biological functions: cancer, organismal injury, and abnormalities (Figure 2). The notable interactions among the differentially expressed proteins included HIF1A, a master transcription factor involved in a multitude of cellular processes in response to hypoxia (Signaling by PTK6 and HIF1A pathway); KDM5A, a lysine specific demethylase 5A, also known as JARID1A or RBP2, that removes lysine 4 of histone H3 (H3K4); and PI3K p85, a glucocorticoid receptor/transcription factor (PI3K) and its catalytic subunit (p85) (Figure 2). Only the notable direct interactions with more than one differentially expressed protein are discussed in detail.

Twelve proteins in network 2 (8 downregulated and 4 upregulated) were identified to be associated with three affected biological functions—metabolic diseases, organismal injury, and abnormalities—with a P-score = 28 (Figure 3). As described in Figure 3 the highlighted or notable interactions included association of differentially expressed proteins with tumor protein 53 (TP53); a tumor suppressor protein, NR3C1; a nuclear receptor subfamily 3 group C member1, Kruppel-like factor 6 (KLF6); a Zinc finger containing transcription factor and ESR1; an estrogen receptor.

Although none of the differentially expressed proteins were common between network 1 and 2, their pathways were related to metabolic diseases, cancer, and organismal injuries. Furthermore, merging both networks (original 11 proteins from network 1 and 12 proteins from network 2), identified convergence pathway with only one cross-interaction protein—p53 (Figure 4).

### 3.3. Hepatotoxicity Report and Identified Irregularities in Liver Function Related to Differentially Expressed Proteins Between HT (Heterozygous LCHAD Mice Without Cancer) and WT Mice

IPA’s core analysis includes “Top tox functions,” which provides a toxicity report of several organs such as the liver. The hepatotoxicity analysis results identified four enriched functional groups exceeding the −log(*p*-value) threshold of 1.25: (1) liver steatosis, (2) glutathione depletion in liver, (3) hepatocellular carcinoma, and (4) liver hyperplasia/hyperproliferation (Figure 5). Downregulated Acyl-CoA dehydrogenase long-chain (ACADL) and methionine adenosyl transferase 1, alpha (MAT1A) were the two differentially expressed proteins associated with liver steatosis. MAT1A was the only protein associated with glutathione quantity in the liver. Two downregulated proteins, N-myc downstream regulator 2 (NDRG2) and indoleamine 2,3-dioxygenase 2 (IDO2), and two upregulated proteins, heat shock protein family A member 9 (HSPA9) and GMP synthase (GMPS) were associated with hepatocellular carcinoma. Eight downregulated proteins Dihydoxylipoamide acetyl transferase (DLAT), catechol-o-methyltransferase (COMT), copper chaperone for superoxide dismutase (CCS), aldehyde dehydrogenase 7 family member A1 (ALDH7A1), nitrilase 1 (NIT1), malate dehydrogenase (MDH1), NDRG2, IDO2 and two upregulated proteins, HSPA9 and GMPS, were associated with liver hyperplasia/hyperproliferation within the hepatocellular carcinoma Tox function.

### 3.4. Differential Expression of p53 and MDM2 in WT, HT (Heterozygous LCHAD Mice Without Cancer) and HT-NC (Heterozygous LCHAD Mice with Non-Cirrhotic Cancer)

The level of p53, which is observed to be the only protein when network 1 and network 2 were merged together and associated with maximum count of differentially expressed proteins (network 1—YWHAE, MDH1, DLAT, PA2G4, NDRG2, and HSPA9; network 2—ACADL, COX5A, ANXA6, SUCLG2, GAPDH, GMPS, and TRAP1), was measured in the livers of WT, HT, and HT-NC mice. To find out how p53 activity was regulated, the level of MDM2 was also measured in the respective samples using SDS-PAGE/Western blot (Figure 6A). Densitometric representation of Western blot was included in Figure 6B (p53) and Figure 6C (MDM2). Significantly decreased p53 (*p* < 0.0001) and increased MDM2 (*p* < 0.0001) expression were noticed in HT-NC liver tissues as compared to WT and HT mice. However, the expression level of p53 is significantly different between WT and HT mice (*p* < 0.001), but that of MDM2 is not significantly different (ns; *p* = 0.9956).

## 4. Discussion

Fatty acid oxidation plays an important role in tumor development [24]. Downregulated HADHA and altered fatty acid β-oxidation were reported by Tanaka et al. in liver cancer patients suggesting the role of HADHA in tumor differentiation [25]. We also reported that defective LCHAD is associated with mitochondrial dysfunction, altered expression of early cancer markers and non-cirrhotic HCC in a mouse model [9]. Both studies suggest HADHA is a tumor suppressor in HCC. In the current investigation, we also observed a significantly decreased expression of HADHA in HCC patients with different stages of disease and different grades of tumor, further supporting the above-mentioned studies.

In the current investigation, two statistically and biologically significant signaling networks (network 1 and network 2) provided a glimpse into potential interactions involving the differentially expressed proteins in HT mice that may result in impaired liver function and irregular mitogenic signaling leading to cancer. Amongst both networks, TP53 in network 2 is the protein with the highest protein interaction count. It is associated with seven differentially expressed proteins (three upregulated—GAPDH, GMPS, and TRAP1—and four downregulated—ACADL, ANXA6, SUCLG2, and COX5A). Interestingly, TP53 was the protein that also cross-interacted with the highest number of differentially expressed proteins from both network 1 and network 2 when merged (network 1—YWHAE, MDH1, DLAT, PA2G4, NDRG2, and HSPA9; network 2—ACADL, COX5A, ANXA6, SUCLG2, GAPDH, GMPS, and TRAP1) (Figure 4). The homeostasis of MDM2-p53 axis is important for normal function of p53 and MDM2. However, both p53 and MDM2 are expressed abnormally in HCC tissues [26]. In LCHAD-defective mouse model, we also noticed abnormal expressions of p53 and MDM2 in HT-NC as compared to WT and HT mouse liver tissues. Increased MDM2 and decreased p53 in HT-NC indicate MDM2-mediated p53 inactivation possibly by ubiquitination. Although the role of p53-MDM2 axis was discussed in many cancers, including cirrhotic HCC, this is the first report to associate this axis with non-cirrhotic HCC.

Apart from TP53, we identified three important interactions in network 1 (Figure 2) that included hypoxia-inducible factor-1A (HIF-1A), PI3K/p85, and KDM5A. The transcription factor, HIF-1A is associated with two downregulated differentially expressed proteins (CCS and CYC1) (Figure 2). Activated HIF1A has been reported in several hepatocellular tumors as a potential circulating marker or target for diagnosis, prognosis and precise treatment of HCC [27,28]. Inhibitors of HIF1A have been used to prevent the recurrence of HCC after radiofrequency ablation (RFA) and high-intensity focused ultrasound (HIFU) ablation [29]. Furthermore, PI3K/p85 is associated with downregulated proteins PA2G4 and DLAT and upregulated HSPA9 (Figure 2). It is a key player in PI3K transduction signaling pathway underlying HCC [30]. Additionally, KDM5A directly interacts with downregulated proteins NIT1, MDH1, and DLAT (Figure 2). It is a lysine-specific demethylase 5A that removes di- and tri-methyl groups from lysine 4 of histone H3 (H3K4) [31]. It acts as a negative regulator of p53 signaling and genetic deletion of this histone demethylase results in upregulation of p53 in multiple cancers and inhibits tumor growth in a p53-miR-34 dependent manner [32].

In network 2 (Figure 3), the four important interactions observed with differentially expressed proteins in HT mice are NR3C1, KLF6, ESR1, and TP53. NR3C1, the glucocorticoid receptor (GR), is associated with three downregulated differentially expressed proteins (AS3MT, COMT, and MAT1A). The loss of GR function in mice leads to steatosis and hepatic tumorigenesis [33]. Recently, the expression of lipid metabolism-associated key genes revealed the prognostic capability of NR3C1 as a potential biomarker for HCC [34]. Further, KLF6 in network 2 is associated with upregulated protein GAPDH and downregulated MCCC1. KLF6 is frequently lost genomically or inactivated by loss of heterozygosity and/or mutation and thus significantly downregulated in cancerous tissues as compared to adjacent non-cancerous tissues [35,36]. Furthermore, estrogen receptor gene 1 (ESR1) in network 2 is associated with upregulated CSAD and GAPDH and downregulated COMT. The sex steroid hormone estrogen’s protective role in HCC has been suggested for a long time [37]. According to Bhat et al. increased expression of ESR1 is prognostic for HCC, protective for overall survival, and more pronounced in women HCC patients [38].

In Tox analysis we identified four enriched functional groups that contained several differentially expressed proteins required for normal liver function and tumor formation. MAT1A, a biological precursor to glutathione, is downregulated in HT mice, thereby leading to decreased glutathione levels. The depletion of MAT1A results in less biosynthesis of S-adenosylmethionine. MAT1A knockout mice model leads to spontaneous steatohepatitis and HCC within an 18-month period [39]. It is important to mention that we also noticed HCC in 13-month-old or older LCHAD mice. Further, ACADL, a mitochondrial enzyme that regulates the primary step of mitochondrial fatty acid oxidation and is thereby responsible for inhibition of HCC growth is downregulated in our defective LCHAD mouse model [40]. In mice, GMPS upregulation has been implicated as an oncogene in HCC development [41]. Interestingly, GMPS is negatively regulated by TP53 and is functionally associated with cellular senescence programming through the TP53 signaling cascade [41]. Other notable proteins expressed differentially, which are associated with HCC and liver hyperplasia/hyperproliferation functional groups include upregulated HSPA9, GMPS and downregulated NDRG2 and IDO2. The inhibition of HCC adhesion, migration, and invasion by NDRG2 is mediated through the regulation of CD24 expression [42]. Wang et al. reported the role of downregulated NDRG2 in the promotion of HCC angiogenesis via VEGFA and its potential use as an anti-angiogenesis target [43]. The dysregulation of all these proteins has been observed in human HCC coupled with a poorer clinical outcome [44,45,46]. Furthermore, downregulated COMT, IDO2, NDRG2, NIT1, ALDH7A1, CCS, and TCA cycle enzymes MDH1 and DLAT, along with upregulated HSPA9 and GMPS are also involved in the hyperplasia/hyperproliferation group. Studies have reported the role of COMT in the development of HCC due to its effect on the biosynthesis of sex hormones estrogen and androgen and their inactivation. Its decreased abundance in male HT mouse can also possibly be attributed to the inactivation of sex hormones [47]. Low expression of COMT and higher methylation of the gene is associated with poor HCC-specific prognosis [48]. Inhibitors of IDO2, an immunosuppressive enzyme, have been implicated in HCC cancer therapy [49,50]. Downregulation of NIT1, a tumor suppressor gene, is reported in HCC specimens as compared to normal liver tissue [51]. Downregulation of MDH1 and DLAT, enzymes of oxidative phosphorylation, are indicative of Warburg-like phenotype of cancer cells and anti-Warburg therapies in combination with sorafenib are the future of advanced HCC therapy [52]. The constant deficiency of CuZnSOD in mice leads to oxidative damage and is associated with HCC in later parts of life. CCS being the chaperone for CuZnSOD, which is downregulated also indicate oxidative stress in the liver tissues [53].

We created a novel knockout mouse model for LCHAD defect that codes for the active site in the exon 15 of the HADHA gene. The human Protein Atlas reveals that HADHA levels are highest in hepatocytes. Hepatocytes, eighty percent of liver cell population, are responsible for energy and fatty acid metabolism with respect to mitochondrial β-oxidation of long chain fatty acids. While HADHA is present in other liver cells, its function and concentration are less central than in hepatocytes. The Human Protein Atlas also reveals that cholangiocytes, another epithelial subtype lining the bile duct, have negligible expression of HADHA compared to hepatocytes. Kupffer cells, which are primarily involved in immune responses and inflammation, are not a main site for the HADHA-driven β-oxidation of fatty acids. However, Kupffer cells can indirectly modulate fatty acid oxidation in nearby hepatocytes [54,55]. Further, hepatic Stellate cells are also not major sites of HADHA-driven fatty acid oxidation and their HADHA levels are therefore minor compared to hepatocytes. During liver injury, Stellate cells activate and transform into myofibroblasts. This process involves significant metabolic reprogramming [56]. However, they are typically involved in liver fatty acid metabolism through fatty acid utilization involving plasmalemma vesicle-associated protein [57]. Thus, in the current investigation we could not decisively draw conclusions about LCHAD-driven regulation of p53/MDM2 axis in a specific subset of liver cells, though it mostly favors hepatocytes over the other cell types. Additionally, in the future, cell-specific LCHAD-defective mouse models are required to strengthen our hypothesis.

In summary, we reported for the first time that TP53 plays a critical role in HCC development in a non-cirrhotic mouse model by interacting with several oncogenes and tumor suppressor genes as identified in IPA. Future studies are required to confirm these associations in patients.

## 5. Conclusions

In conclusion, the analysis provided us with information that p53 plays an important role in HCC-NC and differentially expressed proteins are involved in liver damage, HCC progression, and liver hyperplasia/hyperproliferation. All the notable interactions of network 1 and network 2 are either presented as tumor suppressor/promoter of HCC, serum markers for early HCC diagnosis, or targets for HCC treatment. Therefore, important interactions in network 1 and network 2 further strengthen the idea that defective LCHAD is a novel etiology of HCC. The study indicates that combining the power of network signaling analysis and identifying individual proteins related to liver function provides the necessary following steps into elucidating the potential downstream effects of LCHAD heterozygosity in liver. Validation of the levels of MDM2 and p53 also highlights the role of MDM2-p53 axis in LCHAD-induced MASLD and HCC-NC.

## Figures and Tables

**Figure 1 cancers-17-03241-f001:**
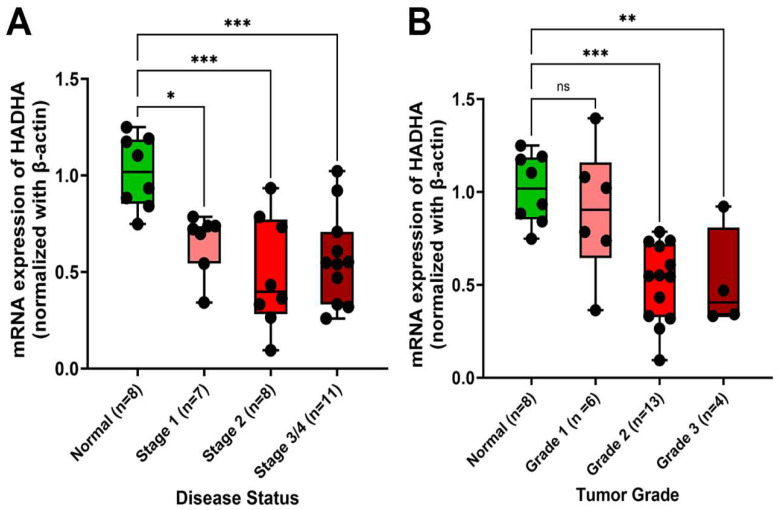
qRT-PCR analysis of HADHA transcript level in HCC patients. Differences in HADHA transcript were observed in patients with disease status from stages 1 to 4 (**A**) and tumor grades from 1 to 3 (**B**) as compared to control (normal) patients. (ns = 0.999; * *p* < 0.01; ** *p* < 0.001; *** *p* < 0.0001).

**Figure 2 cancers-17-03241-f002:**
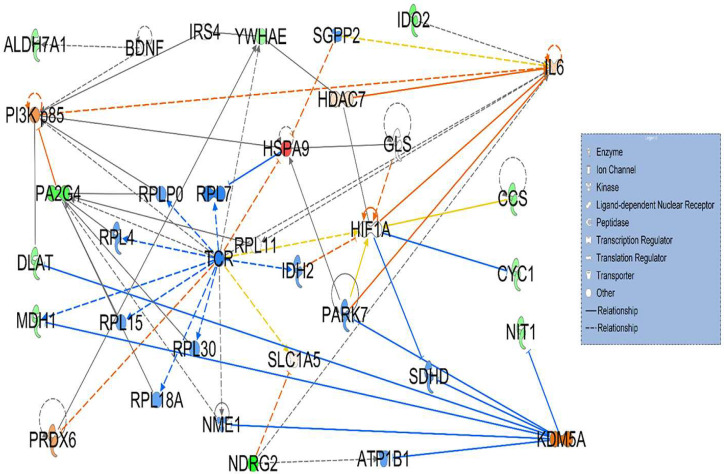
Signaling network 1 was associated with cancer, organismal injury, and abnormalities. Network 1 consisted of 35 nodes and included a total of 11 differentially expressed proteins: 10 downregulated (green) and 1 upregulated (red) in heterozygous LCHAD mice. Note the interactions of differentially expressed proteins CCS and CYC1 with transcription factor HIF1A. Other highlighted protein interactions include MDH1, NIT1, and DLAT interaction with KDM5A and HSPA9, PA2G4, and DLAT interaction with p85, a regulatory subunit of PI3K.

**Figure 3 cancers-17-03241-f003:**
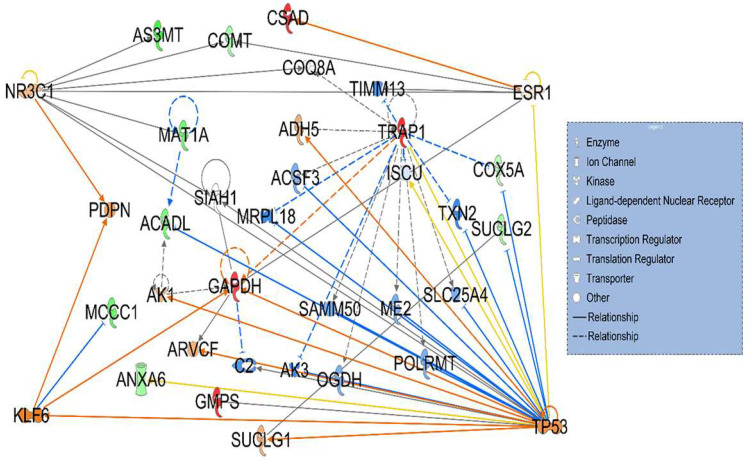
Signaling network 2 was associated with metabolic diseases, organismal injury, and abnormalities. Network 2 consisted of 35 nodes as well and included a total of 12 differentially expressed proteins: 8 downregulated (green) and 4 upregulated (red) in heterozygous LCHAD mice. Note the interactions of differentially expressed proteins AS3MT, COMT, and MAT1A with NR3C1, a key player in glucocorticoid receptor signaling. Also note interactions of GAPDH and MCCC1 with KLF6; CSAD, COMT, and GAPDH with ESR1; and interaction of GAPDH, GMPS, TRAP1, ACADL, ANXA6, SUCLG2, and COX5A with TP53.

**Figure 4 cancers-17-03241-f004:**
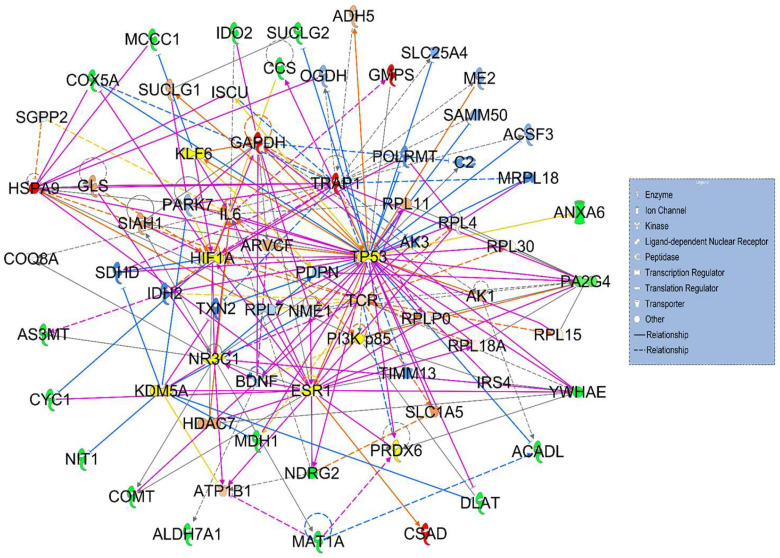
Merging signaling networks 1 and 2. The merged networks consisted of 70 nodes and included a total of 23 differentially expressed proteins: 18 downregulated (green) and 5 upregulated (red) in heterozygous LCHAD mice. Note the predominantly highlighted cross-interactions of differentially expressed proteins from networks 1 and 2 with TP53 that included downregulated proteins YWHAE, MDH1, DLAT, PA2G4, NDRG2, ACADL, COX5A, ANXA6, SUCLG2 (green), and upregulated proteins HSPA9, GAPDH, GMPS, and TRAP1 (red).

**Figure 5 cancers-17-03241-f005:**
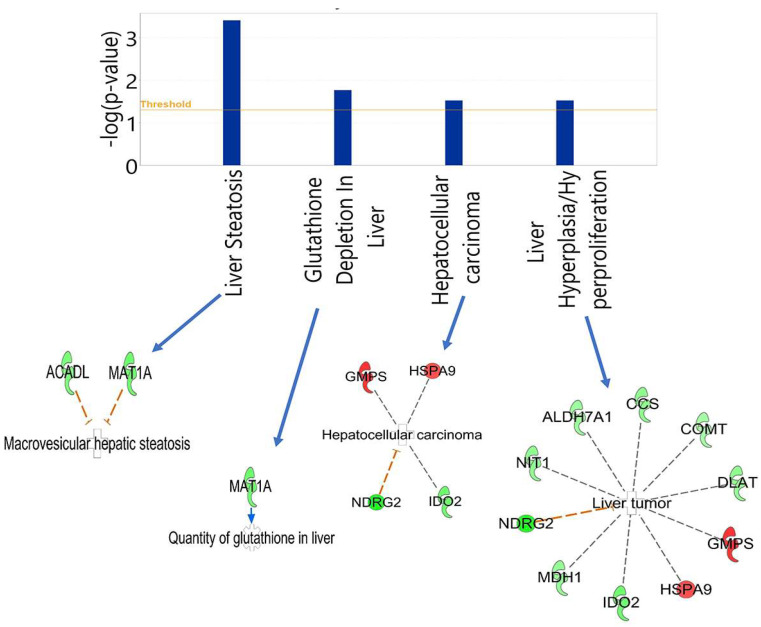
Enriched functional groups from IPA’s hepatotoxicity report. (1) Liver steatosis (microvesicular hepatic steatosis) functional groups [genes = 2]; two downregulated (green) proteins were enriched. (2) Glutathione depletion in liver (quantity of glutathione in liver) functional group [genes = 1]; one downregulated protein (green) was enriched. (3) Hepatocellular carcinoma functional group [genes = 4]; 2 downregulated (green) and two upregulated (red) proteins were enriched. (4) Liver hyperplasia/hyperproliferation (liver tumor) functional group [genes = 10]; 8 downregulated proteins (green) and 2 upregulated proteins (red) were enriched.

**Figure 6 cancers-17-03241-f006:**
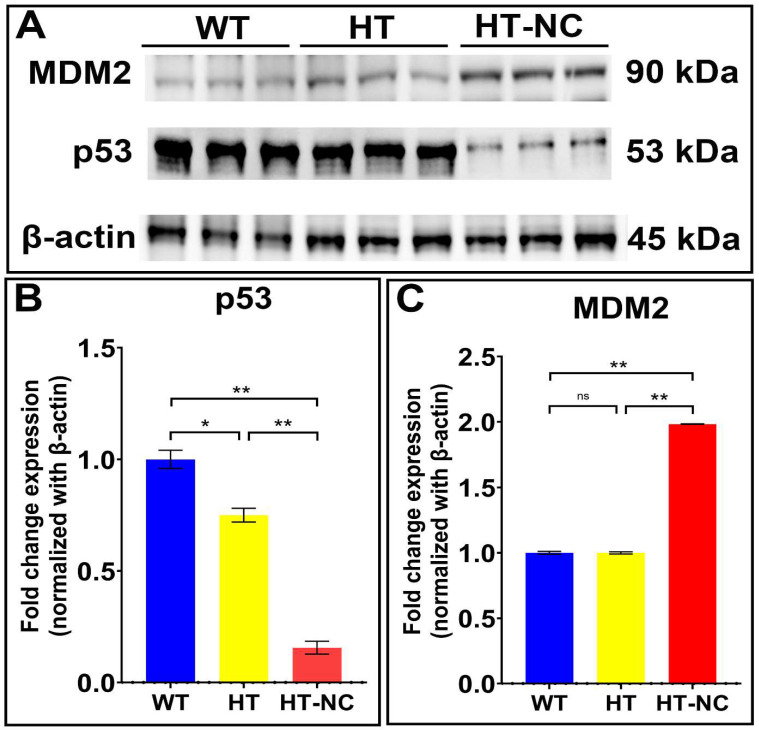
(**A**) Differential expression of p53 and MDM2 in liver tissue of WT (wild type), HT (LCHAD-HT mice with no cancer), HT-NC (LCHAD-HT mice with non-cirrhotic cancer) as noticed in Western blot. Representative Western blot results are shown in the figure. β-actin was used as loading control. (**B**) Densitometric quantification of Western blot revealed significantly reduced expression of p53 in HT mice when compared to WT mice (* *p* < 0.001) and HT-NC mice as compared to both WT and HT mice (** *p* < 0.0001). (**C**) Densitometric quantification of Western blot revealed significantly increased expression of MDM2 in HT-NC as compared to both WT and HT mice (** *p* < 0.0001). However, a non-significant difference (ns; *p* = 0.9956) was noticed between WT and HT mice.

## Data Availability

Data sets will be shared if a formal request has been made. There are no links to publicly archived datasets analyzed or generated during the study.

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
