# Peer review of "Association of TP53 with Defective Long Chain 3-Hydroxy acyl-CoA Dehydrogenase Induced Non-Cirrhotic Hepatocellular Carcinoma"

_cancers, 2025, doi:10.3390/cancers17193241_

Round 1
Reviewer 1 Report
Comments and Suggestions for Authors
In the present study, the authors investigated the biological networks affected in LCHAD-deficient mice that developed non-cirrhotic HCC, based on differentially expressed proteins identified in their previous work. IPA analysis indicated a potential role of the tumor suppressor p53, and Western blot analysis confirmed that p53 was downregulated while its regulator MDM2 was upregulated in the livers of LCHAD-deficient mice. Based on these findings, the authors suggested that defective LCHAD is a novel etiology for HCC.
Although this study may provide new insights into the molecular mechanisms by which LCHAD alteration could contribute to HCC development, the authors’ discussion and hypothesis do not appear to be sufficiently supported by experimental evidence. The conclusions rely heavily on predictions from software analyses without adequate experimental validation. The following concerns should be addressed before the manuscript can be accepted for publication.
(1)
First, there is no information on which cell types are affected by HADHA deficiency, e.g., hepatocytes, immune cells, Kupffer cells, stellate cells, etc. Therefore, although MDM2 upregulation and p53 downregulation were demonstrated in mouse HT-NC samples, it remains unclear in which specific cell types p53 was downregulated. Moreover, given that p53 is frequently downregulated or mutated in HCC and other cancer types, the relationship between p53 downregulation and the development of non-cirrhotic HCC remains uncertain.
(2)
Regarding the terminology for NAFLD and NASH, these terms have been replaced by MASLD and MASH, respectively, several years ago. Please update these terms throughout the manuscript.
Reference: https://pmc.ncbi.nlm.nih.gov/articles/PMC11262457/
(3)
Line 31: “two biological networks”
- Please specify which networks are being referred to.
(4)
Line 138: “Thermocycler (Roche, Madison, WI)”
- Please include the model information, e.g., LightCycler PRO.
(5)
Lines 150–151: “Ingenuity Pathway Analysis (IPA)”
- Please provide the software provider information, e.g., Qiagen.
(6)
Line 169: “Proteins in the whole cell lysates”
- Does this refer to protein concentration? Please clarify.
(7)
Section 3.1. HADHA transcript levels in cDNA array of HCC patients:
- Although the association between HADHA gene alteration and non-cirrhotic HCC was proposed, the samples were not classified into non-cirrhotic and cirrhotic HCC. Therefore, it remains unclear how the HADHA gene is associated with the non-cirrhotic phenotype. Was HADHA unaltered in cirrhotic HCC? Please clarify.
(8)
Section 3.2: “Signaling networks associated with differentially expressed proteins between HT (heterozygous LCHAD mice without cancer) and WT mice”
- TCR (T cell receptor) is expressed in T cells, and IL6 is secreted primarily by Kupffer cells, suggesting that the diagram likely represents correlations across different cell types. In which specific cell types were the differentially expressed proteins expressed? In which cell types might HIF1A and KDM5A contribute to the regulation of these genes?
Since the differentially expressed proteins were not identified at a single-cell level, it is possible that the depicted interactions do not occur within the same cell types in the LCHAD mouse liver.
- In addition, the difference between the solid and dotted lines in the diagrams should be clearly explained. IPA results may include relationships inferred from literature searches based on parameter settings, e.g., co-occurrence of terms within the same document, even when no direct experimental evidence exists.
A more detailed explanation is needed for Figures 2, 3, and 4 in this section.
(9)
Line 267: “IPA’s core analysis”
- Please explain what “core analysis” entails so that readers unfamiliar with IPA can understand the results.
(10)
Lines 293–294: “Level of p53, which is observed to be the only protein common between both the networks and associated with 23 differentially expressed proteins”
- Please clarify what “both networks” refers to. Figure 2 does not include TP53.
- In addition, this description appears to conflict with the statement in lines 379–382: “Amongst both networks, TP53 in Network 2 is the only protein with the highest protein interaction count. It is associated with seven differentially expressed proteins.” Please ensure these descriptions are consistent throughout the manuscript.
(11)
Section 3.4: Differential expression of p53 and MDM2 in WT, HT (heterozygous LCHAD mice without cancer), and HT-NC (heterozygous LCHAD mice with non-cirrhotic cancer)
- Based on Figure 4, ESR1, which encodes the estrogen receptor, also appears to directly or indirectly interact with many of the differentially expressed proteins. The estrogen receptor has been reported to play an important role in preventing liver cancer development in females, contributing to the male predominance of HCC. Please explain why ESR1 was not analyzed in this study.
(12)
Discussion section
- The discussion is somewhat redundant and includes extensive interpretations that are not directly supported by this study. The involvement of KDM5A, ESR1, and HIF1A was not experimentally analyzed and is only predicted by software analysis, making parts of the discussion speculative. It is recommended that the discussion be more closely aligned with the actual experimental findings of this study
Author Response
We thank the reviewer for their valuable insight with meaningful suggestions and corrections. Accordingly, we have made the changes throughout the manuscript.
Reviewer 1
(1)
First, there is no information on which cell types are affected by HADHA deficiency, e.g., hepatocytes, immune cells, Kupffer cells, stellate cells, etc. Therefore, although MDM2 upregulation and p53 downregulation were demonstrated in mouse HT-NC samples, it remains unclear in which specific cell types p53 were downregulated. Moreover, given that p53 is frequently downregulated or mutated in HCC and other cancer types, the relationship between p53 downregulation and the development of non-cirrhotic HCC remains uncertain.
Ans 1. Since entire analysis is done on whole liver lysates prepared from genetic mouse model, specific cell type couldn’t be assigned. Cell type fractionation is a good idea, and we certainly will try to perform such studies, in future, with fresh mice. To confirm the relationship between p53 downregulation and the development of non-cirrhotic HCC, studies are in progress to assess the effect of p53 overexpression by adenoviral vectors in LCHAD defective mice. However, results from such studies are still not available.
(2)
Regarding the terminology for NAFLD and NASH, these terms have been replaced by MASLD and MASH, respectively, several years ago. Please update these terms throughout the manuscript.
Reference: https://pmc.ncbi.nlm.nih.gov/articles/PMC11262457/
Ans 2. As suggested by the reviewer, we have changed nonalcoholic fatty liver disease (NAFLD) to metabolic dysfunction-associated steatotic liver disease (MASLD) and non-alcoholic steatohepatitis (NASH) to metabolic dysfunction associated with steatohepatitis (MASH) throughout the manuscript.
(3)
Line 31: “two biological networks”
- Please specify which networks are being referred to.
Ans 3. “Two biological networks” are now changed to network 1 and network 2 throughout the manuscript.
(4)
Line 138: “Thermocycler (Roche, Madison, WI)”
- Please include the model information, e.g., LightCycler PRO.
Ans 4. Added Thermocycler (LightCycler ®96-Roche, Madison, WI), Line:131
(5)
Lines 150–151: “Ingenuity Pathway Analysis (IPA)”
- Please provide the software provider information, e.g., Qiagen.
Ans 5. Qiagen’s Ingenuity Pathway Analysis (IPA) software version 01-14 was used, Line:143
(6)
Line 169: “Proteins in the whole cell lysates”
- Does this refer to protein concentration? Please clarify.
Ans 6. Protein concentration in the whole cell lysates was determined using BCA protein assay kit, Line:164
(7)
Section 3.1. HADHA transcript levels in cDNA array of HCC patients:
- Although the association between HADHA gene alteration and non-cirrhotic HCC was proposed, the samples were not classified into non-cirrhotic and cirrhotic HCC. Therefore, it remains unclear how the HADHA gene is associated with the non-cirrhotic phenotype. Was HADHA unaltered in cirrhotic HCC? Please clarify.
Ans 7. The array plate had cDNAs from eight normal and twenty-six liver cancer tissues at different stages of carcinogenesis (seven-stage I, eight-stage II and eleven-stage III & IV) and different grades of tumor (six-grade 1, thirteen-grade 2 and four-grade 3). As described by Origene, samples utilized in the study were identified by the pathologist based on the H&E slide, which reflects the tissue block before cDNA extraction. If cirrhosis is identified on the H&E slide, the pathologist incorporates this finding into the diagnosis. Eight samples were classified as "within normal limits" by the pathologist examining the H&E slide, with the composition characterized as “100% Normal”. Twenty-six samples were classified as “Cancers” with the diagnosis determined from the H&E slide. No cirrhosis was reported for the individual tissue samples stored. However, in the surgical pathology report, four patients were identified with cirrhosis in the background: one in stage I, two in stage II and one in stage III. Consequently, most samples used in this study exhibited no cirrhotic background and no evidence of cirrhosis as determined by H&E staining. Line:119-129
(8)
Section 3.2: “Signaling networks associated with differentially expressed proteins between HT (heterozygous LCHAD mice without cancer) and WT mice”
- TCR (T cell receptor) is expressed in T cells, and IL6 is secreted primarily by Kupffer cells, suggesting that the diagram likely represents correlations across different cell types. In which specific cell types were the differentially expressed proteins expressed? In which cell types might HIF1A and KDM5A contribute to the regulation of these genes?
Since the differentially expressed proteins were not identified at a single-cell level, it is possible that the depicted interactions do not occur within the same cell types in the LCHAD mouse liver.
- In addition, the difference between the solid and dotted lines in the diagrams should be clearly explained. IPA results may include relationships inferred from literature searches based on parameter settings, e.g., co-occurrence of terms within the same document, even when no direct experimental evidence exists.
A more detailed explanation is needed for Figures 2, 3, and 4 in this section.
Ans 8.
As explained above, differentially expressed proteins are not identified at a single-cell level, depicted interactions are certainly not happening within the same cell type. Future studies after cell fractionation could identify cell specific interactions. However, such studies must be performed from fresh mice. In figure 2, 3 and 4, solid and dotted lines are explained in figure captions as direct and indirect relationships, respectively.
(9)
Line 267: “IPA’s core analysis”
- Please explain what “core analysis” entails so that readers unfamiliar with IPA can understand the results.
Ans 9. IPA core analysis identifies a series of canonical pathways, diseases and functions or networks relevant for the molecules in the input list. IPA core analysis uses a human-curated comprehensive knowledge base to provide valuable perspective and profound understanding of experimental datasets. Networks of potentially interacting proteins were identified and placed into node-edge diagrams with gene and gene products represented as nodes and biological relationship between two nodes as an edge (line). These biological networks are supported by at least a reference from literature, from a textbook, or from ingenuity pathways knowledge base stored canonical information. Only networks with more than one focus molecule are presented (Black font lines were already there in the Materials and Methods section 2.2). We added few lines. Line:146-148
(10)
Lines 293–294: “Level of p53, which is observed to be the only protein common between both the networks and associated with 23 differentially expressed proteins”
- Please clarify what “both networks” refers to. Figure 2 does not include TP53.
Ans 10 a): We now changed to network 1 and network 2 to make this clearer. Yes, p53 is not there in network 1 (Figure 2) but on convergence of network 1 and network 2, p53 is the only highlighted interaction with maximum number of protein count (Network 1-YWHAE, MDH1, DLAT, PA2G4, NDRG2, and HSPA9; Network 2-ACADL, COX5A, ANXA6, SUCLG2, GAPDH, GMPS, and TRAP1), Line:327-328
- In addition, this description appears to conflict with the statement in lines 379–382: “Amongst both networks, TP53 in Network 2 is the only protein with the highest protein interaction count. It is associated with seven differentially expressed proteins.” Please ensure these descriptions are consistent throughout the manuscript.
Ans 10 b): The statement is true because all other focus molecules we discussed from network 1 (HIF1A, KDM5A, PI3K p85) and network 2 (NR3C1, KLF6, ESR1) interact with either 2,3 or 4 differentially expressed proteins in HT mice. So, the interaction of p53 with GAPDH, GMPS, TRAP1, ACADL, ANXA6, SUCLG2 and COX5A is the highest count in network 2, Line:322-325
Interestingly, TP53 was the only protein that also cross-interacted with maximum count of differentially expressed proteins when both network 1 and network 2 are merged (Network 1-YWHAE, MDH1, DLAT, PA2G4, NDRG2, and HSPA9; Network 2-ACADL, COX5A, ANXA6, SUCLG2, GAPDH, GMPS, and TRAP1) (Figure 4), Line:327-328
(11)
Section 3.4: Differential expression of p53 and MDM2 in WT, HT (heterozygous LCHAD mice without cancer), and HT-NC (heterozygous LCHAD mice with non-cirrhotic cancer)
- Based on Figure 4, ESR1, which encodes the estrogen receptor, also appears to directly or indirectly interact with many of the differentially expressed proteins. The estrogen receptor has been reported to play an important role in preventing liver cancer development in females, contributing to the male predominance of HCC. Please explain why ESR1 was not analyzed in this study.
Ans 11. The reviewer has a very valid point about ESR1 that it should have been analyzed because of its direct interaction with differentially expressed proteins CSAD, COMT, GAPDH. We are certainly hoping to work on the targets like ESR1, KDM5A, HIF1A, and PI3K in future studies. We pursued only p53 because merging of signaling network 1 and network 2 results in p53 as the single protein which interacts with maximum number of differentially expressed proteins from both signaling networks (Network1-YWHAE, MDH1, DLAT, PA2G4, NDRG2, and HspA9; Network 2-ACADL, COX5A, ANXA6, SUCLG2, GAPDH, GMPS, and TRAP1).
(12)
Discussion section
- The discussion is somewhat redundant and includes extensive interpretations that are not directly supported by this study. The involvement of KDM5A, ESR1, and HIF1A was not experimentally analyzed and is only predicted by software analysis, making parts of the discussion speculative. It is recommended that the discussion be more closely aligned with the actual experimental findings of this study
Ans 12. As we discussed above, we are certainly hoping to extend our work to targets like ESR1, KDM5A, HIF1A, and PI3K in future. We wanted to discuss every target and interactions identified in a new non-cirrhotic mouse model, more importantly with TP53 interactions. We have now reduced the discussion to be more concise and not redundant by deleting many lines and associated references (Reduced by 93 lines).

Reviewer 2 Report
Comments and Suggestions for Authors
In this study, the authors demonstrated that p53 signaling is downregulated in LCHAD-deficient mice, underscoring the importance of the anti-oncogenic property of LCHAD.
1. The newly defined term MASLD should be used instead of NAFLD.
2. The Introduction should be more concise. There are too many descriptions that are unrelated to this study, such as HCC etiology (cirrhotic vs. non-cirrhotic), DNA methylation, regionality, and so on. These descriptions should be simplified, though they do not necessarily need to be removed.
3. Materials: How old were the mice used for recovery of liver tissue? Is HT-NC non-cancerous liver tissue?
4. Figure 1: The expression levels of HADHB, TP53, and MDM2 mRNAs should also be determined.
5. Figures 2–5: Because HT mice eventually not developed HCC, these comparisons between WT and HT should not be discussed in the context of liver carcinogenesis unless its expression is confirmed in HT-NC, like p53. For example, HIF1A and PI3K are well-known oncogenes that also play a role in the development of HCC. However, this has not been verified in the LCHAD-deficient mice. The Results and Discussion should be revised accordingly.
Author Response
Reviewer 2
We thank the reviewer for their valuable insight with meaningful suggestions and corrections. Accordingly, we have made the changes throughout the manuscript.
- The newly defined term MASLD should be used instead of NAFLD.
Ans 1. As suggested by the reviewer, we have changed nonalcoholic fatty liver disease (NAFLD) to metabolic dysfunction-associated steatotic liver disease (MASLD) throughout the manuscript.
2. The Introduction should be more concise. There are too many descriptions that are unrelated to this study, such as HCC etiology (cirrhotic vs. non-cirrhotic), DNA methylation, regionality, and so on. These descriptions should be simplified, though they do not necessarily need to be removed.
Ans 2. We have considerably reduced the Introduction and references.
- Materials: How old were the mice used for recovery of liver tissue? Is HT-NC non-cancerous liver tissue?
Ans 3: We have utilized frozen liver samples from previous study. Detailed mice description is given in our previous publication, in brief, as mentioned in introduction, 20% of heterozygous (HT) male mice develop HCC at age 13 months or older without any evidence of fibrosis or cirrhosis (HT-NC) and remaining 80% HT and wild-type (WT) never developed HCC. We have used mice 13 months or older and included age of mice in Materials & Methods, section 2.3, Line: 161
Also, HT-NC is not non-cancerous liver tissue. It is non-cirrhotic cancerous liver tissue (As described in Materials & Methods section 2.3, Line: 162-163.
4. Figure 1: The expression levels of HADHB, TP53, and MDM2 mRNAs should also be determined.
Ans 4. Fresh liver samples are needed to assess mRNA as we haven’t stored liver samples in RNA preserving solution. We will certainly perform these experiments in future.
- Figures 2–5: Because HT mice eventually not developed HCC, these comparisons between WT and HT should not be discussed in the context of liver carcinogenesis unless its expression is confirmed in HT-NC, like p53. For example, HIF1A and PI3K are well-known oncogenes that also play a role in the development of HCC. However, this has not been verified in the LCHAD-deficient mice. The Results and Discussion should be revised accordingly.
Ans 5. As described in our earlier paper (Khare et al, Int J Cancer, 2020, 147;1461-1473), and introduction of this paper, 20% of heterozygous mice develop non-cirrhotic HCC (HT-NC). However, in proteomic (previous study) and IPA analysis (current study), we have used 80% of heterozygous mice which don’t develop cancer (HT). In the current study HT-NC are used only for western blotting experiments for p53 and MDM2 analysis. In previous study, proteins with significant differences between WT and HT mice were identified as early HCC markers (now supplementary Table-1). We validated the expression of these proteins (GAPDH, GMP synthase, Hsp70, TRAP1 and CCS1) by SDS-PAGE in WT, HT, and HT-NC mice in our earlier publication. Thereby, the impairment of long chain fatty acid oxidation because of defective LCHAD (HADHA product) suggests LCHAD as a novel etiology associated with HCC.
As suggested by the reviewer, HIF1A and PI3K should surely be analyzed in these samples. We are certainly hoping to work on the targets like HIF1A, PI3K, KDM5A, and ESR1. We pursued p53 first because merging of signaling network 1 and network 2 results in p53 as the single protein which interacts with maximum number of differentially expressed proteins from both signaling networks (Network1- YWHAE, MDH1, DLAT, PA2G4, NDRG2, and HspA9; Network 2- ACADL, COX5A, ANXA6, SUCLG2, GAPDH, GMPS, and TRAP1).
Round 2
Reviewer 1 Report
Comments and Suggestions for Authors
Most of the concerns raised by this reviewer have been sufficiently addressed or answered. However, a few comments still need to be considered before publication.
Related to Ans 1.:
Data on gene and protein expression in specific cell types are available from public databases such as the Human Protein Atlas (https://www.proteinatlas.org/). Providing information about the cell types that express HADHA may help readers better understand the results and their interpretation in the present study.
Alternatively, the authors could add a statement acknowledging the limitations of the present study, along with a brief note on what is planned for future investigations.
Author Response
Reviewer 1
Question:
Data on gene and protein expression in specific cell types are available from public databases such as the Human Protein Atlas (https://www.proteinatlas.org/). Providing information about the cell types that express HADHA may help readers better understand the results and their interpretation in the present study. (Answer provided by authors)
Alternatively, the authors could add a statement acknowledging the limitations of the present study, along with a brief note on what is planned for future investigations. (Statement added by authors)
Answer:
In "Discussion" (Lines 401-420) we have provided detailed information regarding HADHA expression and any significant role, if any, in fatty acid oxidation in different cell types.
In the current investigation, we could not decisively conclude for LCHAD driven regulation of p53/MDM2 axis in a specific subset of liver cells, though it mostly favors hepatocytes over the other cell types. Further, in future, cell specific LCHAD defect mouse models are required to strengthen our hypothesis.
Reviewer 2 Report
Comments and Suggestions for Authors
The authors have adequately addressed my comments.
I have no further comments.
Author Response
We thank the reviewer for accepting our answers provided in the Round 1. There are no further concerns from this reviewer.